# Hindsight PRIORs for Reward Learning from Human Preferences

**Mudit Verma** [*]
Arizona State University
Tempe, AZ, 85281
muditverma@asu.edu

**Katherine Metcalf**
Apple Inc.
Cupertino, CA, 95014
kmetcalf@apple.com

## Abstract

Preference based Reinforcement Learning (PbRL) removes the need to hand specify a reward function by learning a reward from preference feedback over policy behaviors. Current approaches to PbRL do not address the credit assignment problem inherent in determining which parts of a behavior most contributed to a preference, which result in data intensive approaches and subpar reward functions. We address such limitations by introducing a credit assignment strategy (Hindsight PRIOR) that uses a world model to approximate state importance within a trajectory and then guides rewards to be proportional to state importance through an auxiliary predicted return redistribution objective. Incorporating state importance into reward learning improves the speed of policy learning, overall policy performance, and reward recovery on both locomotion and manipulation tasks. For example, Hindsight PRIOR recovers on average significantly ($p < 0.05$) more reward on MetaWorld (20%) and DMC (15%). The performance gains and our ablations demonstrate the benefits even a simple credit assignment strategy can have on reward learning and that state importance in forward dynamics prediction is a strong proxy for a state's contribution to a preference decision.

## 1 Introduction

Preference-based reinforcement learning (PbRL) learns a policy from preference feedback removing the need to hand specify a reward function. Compared to other methods that avoid hand-specifying a reward function (e.g. imitation learning, advisable RL, and learning from demonstrations), PbRL does not require domain expertise nor the ability to generate examples of desired behavior. Additionally, PbRL can be deployed as human-in-the-loop allowing guidance to adapt on-the-fly to sub-optimal policies, and has shown to be highly effective for complex tasks where reward specification is not feasible (e.g. LLM alignment) Akrour et al. (2011); Ibarz et al. (2018); Lee et al. (2021a); Fernandes et al. (2023); Hejna III & Sadigh (2023); Lee et al. (2023); Korbak et al. (2023); Leike et al. (2018); Ziegler et al. (2019); Ouyang et al. (2022); Zhu et al. (2023). However, existing approaches to PbRL require large amounts of human feedback and are not guaranteed to learn well-aligned reward functions. A reward function is "well-aligned" when policy learned from it is optimal under the target reward function. We address the above limitations by incorporating knowledge about key states into the reward function objective.

Current approaches to learning a reward function from preference feedback do not impose a credit assignment strategy over how the reward function is learned. The reward function is learned such that preferred trajectories have a higher sum of rewards (returns) and consequentially are more likely to be preferred via a cross-entropy objective Christiano et al. (2017). Without imposing a credit assignment strategy to determine the impact of each state on the preference feedback, there are many possible reward functions that assign a higher return to the preferred trajectory. To select between possible reward functions large amounts of preference feedback are required. In the absence of enough preference labelled data, reward selection can become arbitrary, leading to misaligned reward functions. Therefore, we hypothesize that: (H1) guiding reward selection according to state importance will improve reward alignment and decrease the amount of preference feedback required

---

[*]Work during internship at Apple

to learn a well-aligned reward function and (H2) state importance can be approximated as the states that in hindsight are predictive of a behavior's trajectory.

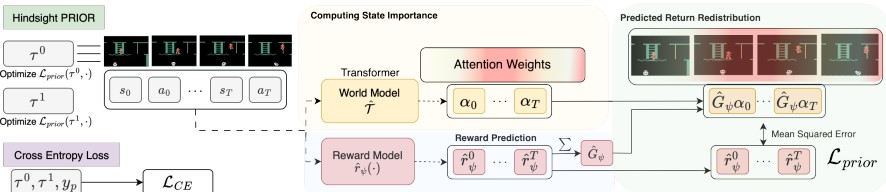

Figure 1: Hindsight PRIOR augments the existing PbRL cross-entropy loss by encouraging the magnitude of a reward to be proportional to the state's importance. Each reward update preference labelled trajectories are passed to a world model $\hat{\mathcal{T}}$ (yellow) and estimated reward $\hat{r}_\psi$ (red), which assign an importance score and a reward (respectively) to each state-action pair. The return $\hat{G}_\psi$ is then applied to the importance scores, which then serve as auxiliary targets for reward learning.

To this end, we introduce *PRIor On Reward* (PRIOR), a PbRL method that guides credit assignment according to estimated state importance. State importance is approximated with an attention-based world model. The reward objective is augmented with state importance as an inductive bias to disambiguate between the possible rewards that explain a preference decision. In contrast to previous work, our contribution mitigates the credit assignment problem, which decreases the amount of feedback needed while improving policy and reward quality. In particular, compared to baselines, Hindsight PRIOR achieves $\geq 80\%$ success rate with as little as half the amount of feedback on MetaWorld and recovers on average significantly ($p < 0.05$) more reward on MetaWorld (20%) and DMC (15%). Additionally, Hindsight PRIOR is more robust in the presence of incorrect feedback.

## 2   RELATED WORK

**PbRL** (Wirth et al., 2017), train RL agents with human preferences on tasks for which reward design is non-trivial and can introduce inexplicable and unwanted behavior (Vamplew et al., 2018; Krakovna et al., 2020). Christiano et al. (2017) extended PbRL to Deep RL and PEBBLE (Lee et al., 2021a) incorporated unsupervised pre-training, reward relabelling, and offline RL to reduce sample complexity. Subsequent works extended PEBBLE by incorporating pseudolabelling into the reward learning process (Park et al., 2022), guiding exploration with reward uncertainty (Liang et al., 2022), and monitoring Q-function performance on the preference feedback (Liu et al., 2022).

Kim et al. (2023) attempts to address the credit assignment problem by assuming that the preference feedback is based on a weighted sum of rewards and use a modified transformer architecture to assign rewards and weights to each state-action pair. However, introducing a transformer-based reward function increases reward complexity compared to earlier work and Hindsight PRIOR as well as tying the reward model to a specific architecture. While Hindsight PRIOR also uses a transformer architecture, it is independent of the reward architecture. Additionally, Kim et al. (2023) has not been extended to online RL.

**Learning World Models:**  Reinforcement Learning, especially model-based RL, leverage learned world models for tasks such as planning (Allen & Koomen, 1983; Hafner et al., 2019b;a), data augmentation (Gu et al., 2016; Ball et al., 2021), uncertainty estimation (Feinberg et al., 2018; Kalweit & Boedecker, 2017), and exploration (Ladosz et al., 2022). In this work we learn a world model and use it to estimate the importance of state-action pairs. While Hindsight PRIOR can use any transformer-based world-model, we use the current state of the art in terms of sample complexity, Transformer-based World Models (TWM) (Robine et al., 2023). To our knowledge, existing work has not incorporated a world model in reward learning from preferences.

**Feature Importance:** Many methods exist to estimate the importance of different parts of an input to the model decision-making process. Some popular methods include gradient / saliency based approaches (Greydanus et al., 2018; Selvaraju et al., 2017; Simonyan et al., 2013; Weitkamp et al., 2019) and self-attention based methods Ras et al. (2022); Wiegreffe & Pinter (2019); Vashishth et al. (2019). Self-attention based methods have been used for video summarization and extraction of key

frames (Feng et al., 2020; Bilkhu et al., 2019; Apostolidis et al., 2021; Liu et al., 2019). Given our use of TWM,, we use a self-attention map based method.

**Credit Assignment:** Credit assignment challenges typically stem from sparse rewards and large state spaces and solutions aim to boost policy learning (Ke et al., 2018; Goyal et al., 2018; Ferret et al., 2020) Past works like Goyal et al. (2018) have learned a backward dynamics model to sample states that could have led to the current state and Ferret et al. (2020) equips a non-autoregressive sequence model to reconstruct a reward function and utilizes model attention for credit assignment. Return redistribution is another credit assignment solution that redistribute the ground-truth, non-stationary reward signal in order to denisfy the emitted reward signals (Ren et al., 2021; Arjona-Medina et al., 2019; Patil et al., 2020). This is in contrast to PbRL where predicted rewards are dense to begin with. We adapt the idea of return redistribution for PbRL by redistributing the predicted returns and is discussed in Section 4.3.

## 3 PREFERENCE-BASED REINFORCEMENT LEARNING

To learn a policy with preference-based reinforcement learning (PbRL), the policy $\pi_\phi$ executes an action $a_t$ at each time step $t$ in environment $\mathcal{E}$ based on its observation $o_t$ of the environment state $s_t$. For each action the environment $\mathcal{E}$ transitions to a new state $s_{t+1}$ according to transition function $\mathcal{T}$ and emits a reward signal $\hat{r}_t = \hat{r}_\psi(s_t, a_t)$. The policy $\pi_\phi$ is trained to take actions that maximize the expected discounted return $\hat{G}_\psi = \sum_t \gamma \hat{r}_\psi(s_t, a_t)$. The reward $\hat{r}_\psi(\cdot)$ is trained to approximate the human's target reward function $\bar{r}_\psi(\cdot)$.

To learn $\hat{r}_\psi(\cdot)$ a dataset $\mathcal{D}$ of preference triplets $(\tau_0, \tau_1, y_p)$ is collected from a teacher (human or synthetic) over the course of policy training. The preference label $y_p$ indicates which, if any, of the two trajectory segments $\tau_0$ or $\tau_1$ with length $l$ has a higher (discounted) return $G_\psi$ under the target reward function $\bar{r}_\psi(\cdot)$. Following Park et al. (2022) and Lee et al. (2021a), feedback is solicited every $K$ steps of policy training for the $M$ *maximally informative* trajectories pairs $(\tau_0, \tau_1)$ (e.g. pairs with the largest $\hat{r}_\psi(\cdot)$ uncertainty).

Given a preference dataset $\mathcal{D}$, $\hat{r}_\psi(\cdot)$ is learned such that preferred trajectories have higher predicted returns $\hat{G}_\psi$ than dispreferred trajectories. Using the Bradley-Terry model (Bradley & Terry, 1952), predicted trajectory returns are used to compute the probability that one trajectory is preferred over the other $P_\psi$:

$$P_\psi[\tau^0 \succ \tau^1] = \frac{\exp \sum_t \hat{r}_\psi(s_t^0, a_t^0)}{\sum_{i \in \{0,1\}} \exp \sum_t \hat{r}_\psi(s_t^i, a_t^i)}, \tag{1}$$

where $\tau_0$ is preferred over $\tau_1$. The probability estimate $P_\psi$ is then used to compute and minimize the cross-entropy between the predicted and the true preference labels:

$$\mathcal{L}_{CE} = \mathop{-\mathbb{E}}_{(\tau_0, \tau_1, y_p) \sim \mathcal{D}} [y_p(0) \log P_\psi[\tau_0 \succ \tau_1] + y_p(1) \log P_\psi[\tau_1 \succ \tau_0]]. \tag{2}$$

The reward function $\hat{r}_\psi$ is learned over the course of policy $\pi_\phi$ training by iterating between updating $\pi_\phi$ according to the current estimate of $\bar{r}_\psi$ and updating $\hat{r}_\psi$ on $\mathcal{D}$, which is grown by $M$ preference triplets sampled from $\pi_\phi$'s experience replay buffer $\mathcal{B}$ for each $\hat{r}_\psi$ update. To avoid training $\pi_\phi$ on a completely random $\hat{r}_\psi$ at the start of training, $\pi_\phi$ explores the environment to populate $\mathcal{D}$ with an initial set of trajectories following either a random policy or during an intrinsically motivated pre-training period Christiano et al. (2017); Lee et al. (2021a).

## 4 HINDSIGHT PRIORS

PbRL relies on learning a high-quality reward function that generalizes and quickly adapts in a few-shot manner to unseen portions of the environment, and given its human in the loop nature, reducing the amount of preference feedback is vital. To learn the reward function $\hat{r}_\psi(s_t, a_t)$, trajectory-level feedback is provided and then is distributed to each of the trajectory's states-action pairs. Given two trajectories, a return per trajectory $(\hat{G}_\psi^0, \hat{G}_\psi^1)$, and a preference label, many reward functions assign a higher return for preferred trajectories, but do not align with the target reward function $\bar{r}_\psi(s_t, a_t)$ on unseen data. With a large enough dataset, a $\hat{r}_\psi(\cdot)$ that aligns with human preferences in all portions of the environment can be learned. However, given a set of reward functions $\hat{r}_\psi(\cdot)$,

each of which conforms to the preference dataset, a reward function will be arbitrarily selected in the absence of additional information or constraints. From insufficient preference feedback, the selected $\hat{r}_\psi(\cdot)$ is likely to represent a local minimum with respect to previously unseen trajectories, where the assigned returns $\hat{R}_\psi$ are correct, but the distribution of rewards within trajectories are incorrect. Incorrectly assigning rewards at the state-action level, or incorrectly solving the credit assignment problem, leads to reward functions that do not generalize outside of the preference dataset, resulting in suboptimal policies relative to the target reward function. Thus, we address the credit assignment problem and guide reward distribution within a trajectory through an auxiliary objective that provides a prior on state-action pair values computed after the trajectory has been observed (in hindsight).

The priors on state-action values are identified by answering the following question, "now that I have seen what happened, which state-action pairs best summarize what happened in the given trajectory?" We consider the states that summarize a trajectory to be those that are most predictive of future state-action pairs. The most predictive states are then used as a proxy for the most important states.The use of summarizing state-action pairs is motivated by previous work demonstrating that people have selective attention when evaluating a behavior – they attend only to the state-action pairs necessary to provide the evaluation (Desimone & Duncan, 1995; Bundesen, 1990; Ke et al., 2018). We therefore assign greater credit to those states that were likely to have been attended to and therefore influenced the preference feedback. As summarizing states are those that are predictive of future state-action pairs, we identify them using an attention-based forward dynamics model, where state-action pair importance is proportional to their weight in the attention layers. For example, in Figure 1 the important states (highlighted in red) identified from an action sequences in Montezuma's Review are those where the agent lines up to leap from the platform.

### 4.1 Approximating State Importance with Forward Dynamics

An attention-based forward dynamics model (Figure 1 yellow) is used to identify important (summarizing) states and address the PbRL credit assignment problem. The states that are key for a forward dynamics model to predict the future are assumed to be similar to those a human evaluator would use to predict future states, and thus summarize a trajectory. We use the attention layers in an attention-based forward dynamics model to approximate human attention and guide how feedback credit is distributed across a trajectory. In similar vein as Harutyunyan et al. (2019)'s State Conditioned Hindsight Credit Assignment, we consider the importance of a state in a trajectory given that a future state was reached.

World models have played a large role in model-based reinforcement learning. Given the power that recent work have shown them to convey in reinforcement learning (Manchin et al., 2019; Hafner et al., 2019a; Hu et al., 2019), we use world modelling techniques to learn an attention-based forward dynamics model. For a world model $\hat{\mathcal{T}}$ to identify important states and approximate human attention, it must have two characteristics. First, it must model environment dynamics and be able to predict the next future state $\hat{s}_T$ given a history of state-action pairs $\tau_{[1:T-1]}$: $\hat{\mathcal{T}}(\tau_{[1:T-1]}) = \hat{s}_T$. Second, it must expose a mechanism to compute state-action importance $\alpha_{[1:T-1]}$ vector over a given trajectory segment $\tau_{[1:T-1]}$ when performing the next-state prediction: $\hat{\mathcal{T}}(\tau_{[1:T-1]}, \hat{s}_T) = \alpha_{[1:T-1]}$. *Transformer based World Models* (TWM) Robine et al. (2023) meets both requirements in addition to being sample efficient (Robine et al., 2023; Micheli et al., 2023).

TWM is a Transformer XL based auto-regressive dynamics model $\hat{\mathcal{T}}$ that predicts the reward $\hat{r}_t$, discount factor $\hat{\gamma}_t$, and (latent) next state $\hat{z}_{t+1}$ given a history of state-action pairs ($\mathcal{W}(\tau_{[1:h]}) = s_h$). In the PbRL paradigm, predicting a transition's reward $r_t$ is impractical as the reward function $\hat{r}_\psi$ is learned in conjunction with the world model. Therefore, we adapt TWM by removing the reward and discount heads, and use the observation and latent state models:

1. Observation Encoder and Decoder: $z_t \sim p_\mu(z_t|o_t)$; $\hat{o}_t \sim p_\mu(\hat{o}_t|z_t)$
2. Aggregation and Latent State Predictor: $h_t = f_\omega(z_{[1:t]}, a_{[1:t]})$; $\hat{z}_{t+1} \sim p_\omega(\hat{z}_{t+1}|h_t)$

Consequentially, the loss function for the dynamics model is updated as follows, where $H$ is the cross entropy between the predicted and true latent next states:

$$\mathcal{L}_\omega^{\text{Dyn.}} = \mathbb{E}[\sum_{t=1}^{T} H(p_\mu(z_{t+1}|o_{t+1}), p_\omega(\hat{z}_{t+1})|h_t)]. \tag{3}$$

The *Latent State Predictor* is a transformer responsible for predicting the forward dynamics given the trajectory history, and is therefore responsible for approximating state-action importance. For a description of the latent state predictor and its architecture, specifically the parts that allow us to extract state importance, see Appendix C.2.

The world model is learned over the course of policy $\pi_\phi$ and reward $\hat{r}_\psi$ training. The observation encoder's and decoder's weights $\mu$ are trained during $\pi_\phi$'s exploration period to initially populate $\mathcal{D}$, and then frozen for the remainder of $\pi_\phi$ and $\hat{r}_\psi$ training. The weights of the dynamics model $\omega$ are trained during $\pi_\phi$'s exploration phase and then updated every $j$ steps of policy training from the same replay buffer $\mathcal{B}$ the preference queries are sampled from. Using $\mathcal{B}$ removes the need to sample additional transitions or trajectories for the purpose of world model learning.

## 4.2 COMPUTING THE HINDSIGHT PRIORs

The use of a transformer-based *Latent State Predictor* provides approximations of state importance in the form of attention weights (our second requirement in Section 4.1). When updating $\hat{r}_\psi$ the attention weights for each trajectory $\tau$ in the collected preference triplets $(\tau_0, \tau_1, y_p) \in \mathcal{D}$ are computed by passing $\tau$ to the Transformer XL model $\hat{\mathcal{T}}$ (Figure 1 yellow). The transformer uses a multi-headed, multi-layer attention mechanism, where $H$ is the number of attention heads, $L$ the number of layers, and $attn_t^l = (attn_{s_t}^l, attn_{a_t}^l) \in \mathcal{A}^{2T \times L}$ the attention weights of the $l$-th layer for state-action pair $(s_t, a_t) \in \tau_{1:T}$. The matrix $\mathcal{A}$ denotes the attention distribution in predicting the next state $\hat{z}_{T+1} = \hat{\mathcal{T}}(\tau)$ across all sequence timesteps and attention layers. The hindsight PRIOR (importance) $\alpha_t$ for a given state-action pair $(s_t, a_t)$ is estimated as the mean across layers $L$ at timestep $t$, $\alpha_t = 1/L \sum_{l=1}^{L} attn_t^l$.

## 4.3 REWARD REDISTRIBUTION AND CONSTRUCTING THE HINDSIGHT PRIOR LOSS

To guide reward function learning according to state-action pair importance, the attention maps $\mathcal{A}$ from $\hat{\mathcal{T}}$ are incorporated into the reward learning objective as redistribution guidance (Figure 1 orange). The attention map does not form a reward target, as state-action importance for predicting future states does not equate absolute value in the target reward function, therefore return redistribution (Arjona-Medina et al., 2019), a strategy typically used to address the challenge of delayed returns in reinforcement learning, is used to align reward assignment with state-action importance.

Return redistribution addresses the challenge of delayed returns by redistributing a trajectory segment's return among its constituent state-action pairs. The return redistribution use case in existing work (Arjona-Medina et al., 2019; Ren et al., 2021; Patil et al., 2020) relied on known and typically stationary, but sparse, rewards. In PbRL, while the learned reward function is dense, the feedback used to learn it occurs at the end of a trajectory and therefore is delayed and sparse. Therefore, to align rewards with estimated state importance, we introduce *predicted* return $\hat{G}_\psi$ redistribution to obtain state-action pair importance conditioned reward targets for a given trajectory $\tau$, where $\hat{G}_\psi = \sum_t^T \hat{r}_\psi(\tau_t)$.

To obtain the reward targets for each trajectory $\tau$ in a preference triplet $(\tau_0, \tau_1, y_p)$, the predicted return $\hat{G}_\psi$ is computed (Figure 1 red), the attention map $\mathcal{A}(\tau) \sim \hat{\mathcal{T}}(\tau)$ is extracted from the world model (Figure 1 yellow), and the mean attention value per state-action pair is taken over layers $\alpha = \frac{1}{L} \sum_{l=1}^{L} (attn_{s_t}^l + attn_{a_t}^l)$. Reward value targets are then estimated by redistributing the predicted return $\hat{G}_\psi$ according to $\alpha$ to obtain $\mathbf{r}_{target} = \alpha \odot \hat{G}_\psi$, where $\alpha$ is a vector with length $|\tau|$ and $\hat{G}_\psi$ a scalar (Figure 1 orange). The state-action pair importance conditioned reward targets $\mathbf{r}_{target}$ are incorporated into reward learning via an auxiliary mean squared error loss between the predicted rewards $\hat{\mathbf{r}}_\psi = [\hat{r}_\psi(s_1, a_1), \hat{r}_\psi(s_2, a_2), ..., \hat{r}_\psi(s_T, a_T)]$ and $\mathbf{r}_{target}$:

$$\mathcal{L}_{prior} = MSE(\hat{\mathbf{r}}_\psi, \mathbf{r}_{target}). \tag{4}$$

The PbRL objective $\mathcal{L}_{CE}$ (Equation 2) is modified to be a linear combination of the proposed hindsight PRIOR loss $\mathcal{L}_{prior}$ to guide reward learning with both preference feedback and estimated

state-action importance:

$$\mathcal{L}_{pbrl}(\mathcal{D}) = \frac{1}{|\mathcal{D}|} \sum_{i=1}^{|\mathcal{D}|} \mathcal{L}_{CE}(\mathcal{D}_i) + \lambda * \mathcal{L}_{prior}(\mathcal{D}_i), \tag{5}$$

where $\lambda$ is a constant to ensure $\mathcal{L}_{CE}$ and $\mathcal{L}_{prior}$ are on the same scale.

## 5 EMPIRICAL EVALUATION

We evaluate the benefits of Hindsight PRIOR on the Deep Mind Control (DMC) Suite locomotion (Tunyasuvunakool et al., 2020) and MetaWorld control (Yu et al., 2020) tasks, compare against baselines (Lee et al., 2021a; Park et al., 2022; Liu et al., 2022; Liang et al., 2022), and ablate over Hindsight PRIOR's contributions. Following our baselines, tasks with hand-coded rewards are used to assess algorithm performance. The hand-coded rewards serve as the target reward functions (used by human in the loop) and are used to assign synthetic preference feedback (trajectories with the higher return are preferred). Therefore, PbRL policy performance is measure and compared according to how well and how quickly the target reward function is maximized. Additionally, a SAC (Haarnoja et al., 2018) policy is trained on the target reward function to provide a reasonable reference point for PbRL performance. Each PbRL method is compared to SAC using mean normalized return for DMC Lee et al. (2021b) and mean normalized success rate for MetaWorld. See Appendix F for the equations. For each comparison against baselines, mean (+standard deviation) policy learning curves and normalized returns are reported over 5 random seeds (see Appendix E). From the learning curves and normalized scores, feedback sample efficiency, environment interaction sample efficiency, and reward recovery are compared between Hindsight PRIOR and baselines.

While using synthetic feedback allows us to directly compare between the target $\bar{r}_\psi$ and learned $\hat{r}_\psi$ reward functions, humans do not always select the trajectory that maximizes the target reward function. Occasionally, humans will mislabel a trajectory pair and flip the preference ordering. Therefore, we evaluate Hindsight PRIOR and PEBBLE (the backbone algorithm for Hindsight PRIOR and the baselines) using a synthetic feedback labeller that provides incorrect feedback on a percentage (10%, 20%, 40%) of the preference triplets (mistake labeller from (Lee et al., 2021b)).

To better understand Hindsight PRIOR's performance gains over baselines (Section 5.1), we answer the following questions in Section 5.2:

- (Q1) Is it the use of a return redistribution strategy versus Hindsight PRIOR's specific strategy (guiding return redistribution according to state importance) that leads to the performance improvements?
- (Q2) Do the performance gains stem from incorporating environment dynamics?
- (Q3) What types of states does TWM identify as important?

and to verify that the incorporation of the world model does not negatively impact PbRL capabilities, we answer the following in Section 5.3:

- (Q4) Does Hindsight PRIOR's scale to longer trajectories in the preference triplets?
- (Q5) Does combining Hindsight PRIOR with a complementary baseline improve performance?
- (Q6) Does Hindsight PRIOR allow for the removal of preference feedback?

Hindsight PRIOR and all baselines extend PEBBLE as their underlying PbRL algorithm. The policy takes random actions for the first 1k steps of policy training and then trains with an intrinsically-motivated reward (as suggested by Lee et al. (2021a)) for 9k steps. The experimental set up and task configurations are selected following Park et al. (2022) which is the existing state of the art method. Algorithm-specific hyper-parameters match those used by the corresponding paper and hyper-parameters determining feedback schedules and amounts match those used in Park et al. (2022) (see Appendix E).

### 5.1 COMPARING AGAINST PbRL BASELINES

Figure 2 and Table 1 compare the performance of Hindsight PRIOR to PEBBLE, SURF Park et al. (2022), RUNE Liang et al. (2022), and MRN Liu et al. (2022) with perfect feedback. The amount

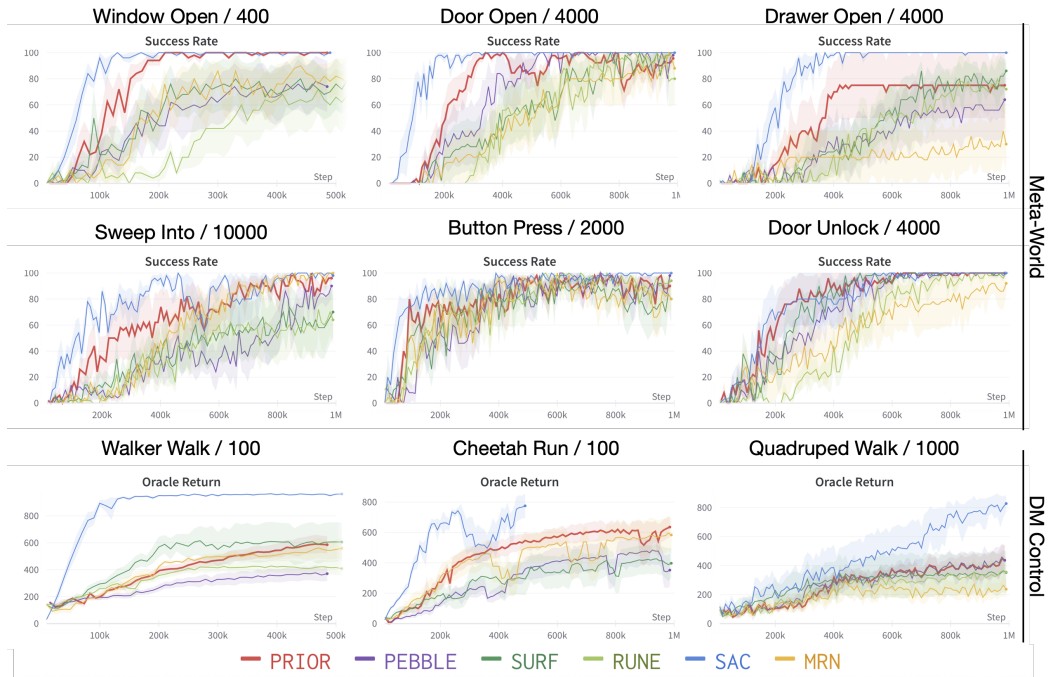

Figure 2: PbRL and SAC policy learning curves for six MetaWorld (top and middle rows) and three DMC (bottom row) tasks. Each experiment is specified as: task / feedback amount.

Table 1: Mean (± variance) normalized success rates (MetaWorld) and normalized returns (DMC) across tasks.

| Suite | PRIOR | PEBBLE | SURF | RUNE | MRN |
|---|---|---|---|---|---|
| MetaWorld | **0.66± 0.002** | 0.56 ± 0.007 | 0.56 ± 0.004 | 0.53 ± 0.009 | 0.55 ± 0.005 |
| DMC | **0.59± 0.003** | 0.48 ± 0.018 | 0.55 ± 0.019 | 0.49 ± 0.007 | 0.53 ± 0.002 |

of feedback is held fixed across methods for a given task and is provided every 5k steps of policy training (X-axis), therefore learning curve performance in Figure 2 relative to the number of policy steps indicates both reward and policy sample complexity. For example, at policy step 30k for `walker-walk`, the preference dataset contains 10 preference triplets and 20 at 50k steps. Table 1 reports the mean normalized return and success rate for each algorithm across tasks and shows that Hindsight PRIOR has the best overall performance across tasks. A two-tailed paired t-test with dependent means was performed over the normalized returns and success rates to determine that Hindsight PRIOR's performance gains are statistically significant. (Appendix F for t and p-scores. Task specific normalized returns and success rates are reported in Appendix F).

For all tasks, Hindsight PRIOR matches or exceeds baseline performance, and for all except `quadruped-walk`, either converges to a higher performance point (e.g. 100% versus 80% success rate on `window-open`) or requires significantly less preference labels to achieve the same performance point (e.g. 100% success rate at $\sim 350$k policy steps versus $\sim 550$k for `door-open`). The results suggest that Hindsight PRIOR's credit assignment strategy improves PbRL beyond guiding exploration with reward uncertainty (Liang et al., 2022), increasing the amount of preference feedback through pseudo-labelling (Park et al., 2022), and incorporating information about policy performance in reward learning (Liu et al., 2022).

Figure 3 (returns left and success rates center) shows the performance differences for PEBBLE (Lee et al., 2021a) and Hindsight PRIOR on `window-open` across different amounts of preference feedback mistakes. The mistake amounts are percentages of the maximum feedback amount, specifically 0% (perfect labeller), 10%, 20%, and 40%. We compare against PEBBLE, because it has compara-

ble performance to the baselines (Figure 2 and Table 1) and is the underlying PbRL algorithm. For all mistake amount conditions, Hindsight PRIOR outperforms PEBBLE. Furthermore, Hindsight PRIOR trained on a dataset with 20% labelling errors beats the performance of PEBBLE with no labelling errors. The results suggest that the inclusion of a credit assignment strategy, specifically one guided by estimated state importance, makes reward and policy learning more robust to preference feedback labelling errors.

## 5.2 UNDERSTANDING THE PERFORMANCE GAINS

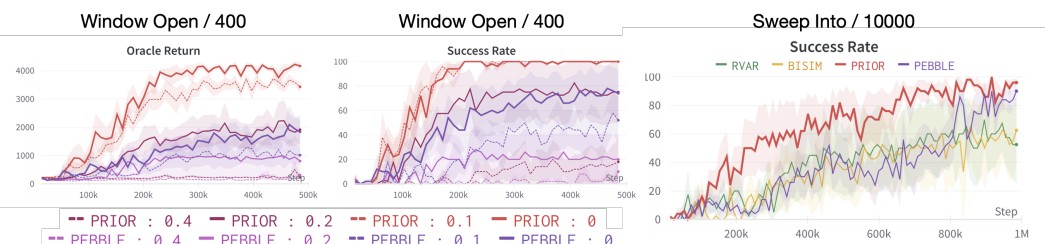

Figure 3: PbRL learning curves over different labelling mistake amounts (left & center : purple & pink for PEBBLE and red & magenta for PRIOR), and different methods for return distribution and dynamics-aware rewards (right).

In order to better understand sources of Hindsight PRIOR's performance gains we evaluate the importance of the state-importance guided return redistribution strategy by comparing against different redistribution strategies (Q1), assess the impact of Hindsight PRIOR making reward learning dynamics aware by replacing $L_{prior}$ with an adapted bisimulation objective (Kemertas & Aumentado-Armstrong, 2021) (Q2), and qualitatively assess what the world model $\hat{\mathcal{T}}$ identifies as important states (Q3). The results show the benefits of the forward dynamics based state-importance redistribution strategy, demonstrate that Hindsight PRIOR's contributions extend beyond making reward learning dynamics aware, and that $\hat{\mathcal{T}}$'s attention weight identify reasonable state-action as important.

We compare against PEBBLE as it has comparable performance to the baselines (Figure 2 and Table 1) and is the underlying PbRL algorithm for all baselines.

**Redistribution Strategy (Q1):** Hindsight PRIOR's redistribution strategy is compared against an uninformed return redistribution strategy, using the mean attention weights $\alpha$ serve as the reward targets (RVAR). The uniform strategy corresponds to assigning uniform importance to each state-action pair in a trajectory and each state-action pair is assumed to equally contribute to the preference feedback. The uniform strategy adapts Ren et al. (2021)(RRD) to obtain the reward target $R_{target}$ by setting $\alpha_t = \frac{1}{T}$. Figure 3 (right - green) shows that while uniform predicted return redistribution is on par with PEBBLE (and in some cases better, see Appendix G.1), Hindsight PRIOR is superior in feedback and environment sample efficiency.

Given Hindsigh PRIOR's performance relative to a uniform redistribution strategy, we amplify Hindsight PRIOR's attention weights through a min-max normalization of the attention map followed by a softmax (NRP). Amplifying the attention map moves it further from the uniform redistribution strategy and potentially improves it. However, Hindsight PRIOR and NPR have comparable performance (Figure 6 in Appendix G.3) showing that explicitly discouraging a uniform redistribution strategy is not necessary.

**Dynamics Aware Reward Learning (Q2):** While Hindsight PRIOR does not directly use the forward dynamics of the world model $\hat{\mathcal{T}}$, knowledge of transition dynamics influence how the reward function is learned. Therefore, we assess the contribution of dynamics-aware reward learning in the absence of a return redistribution credit assignment strategy. To incorporate dynamics, a bisimulation-metric representation learning objective, which has been used as a data-efficient approach for policy learning, is incorporated into reward learning. See Appendix G.2 for details on incorporating the bisimulation auxiliary encoder loss Kemertas & Aumentado-Armstrong (2021) into Hindsight PRIOR.

The results show that making reward learning dynamics aware improves policy learning (Figure 3 (right-yellow)) compared to PEBBLE,but *not* compared to Hindsight PRIOR. Therefore, while incorporating of environment dynamics into reward learning explains part of Hindsight PRIOR's performance gains, it does not explain all of the performance gains highlighting the importance of Hindsight PRIOR's credit assignment strategy.

**Examining Important States (Q3):** Fig. 1 shows the attention over a trajectory snippet from Montezuma's Revenge (analysis in App. I). In our qualitative experiments with discrete domains of Atari Brockman et al. (2016) and control based domains of Metaworld Yu et al. (2020) we found a significant overlap between important states for future state prediction and underlying task.

## 5.3 ASSESSING SCALABILITY AND COMPATIBILITY

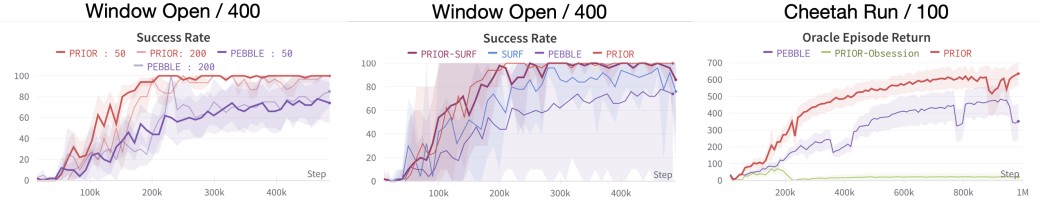

Figure 4: Learning curves evaluating different trajectory lengths (left), combining Hindsight PRIOR with SURF (center), and removing the influence of preference feedback (right).

**Scalability (Q4):** Since Hindsight PRIOR subroutines a forward dynamics model to obtain the attention map $\mathcal{A}$ we evaluate whether it can identify important states in longer trajectories that provide more context for human evaluators. Figure 4 (left) and Appendix H show that, following similar trends as PEBBLE, Hindsight PRIOR's performance is consistent given a 4x increase in trajectory length (50 versus 200 query length).

**Combining with PEBBLE Extensions (Q5):** We investigate the benefits of Hindsight PRIOR when used in parallel with another sample-efficient PbRL techniques, like SURF (Park et al., 2022). Figure 4 (center) shows combining Hindsight PRIOR with SURF (Park et al., 2022) improves policy performance relative to PEBBLE and SURF, but provides no real gain relative to Hindsight PRIOR alone.

**Removing Preference Feedback (Q6):** The results in Figure 4 (right) show the impact of making $\lambda$ very large (green) in Equation 5 resulting in a reward function that is learned solely from $\mathcal{L}_{prior}$. The inability of Hindsight PRIOR to learn anything with a very large $\lambda$ verifies that focusing the reward signal around important states is not sufficient for policy learning.

## 6 CONCLUSION

We have presented Hindsight PRIOR, a novel technique to guide credit-assignment during reward learning in PbRL that significantly improves both policy performance and learning speed by incorporating state importance into reward learning. We use the attention weights of a transformer-based world model to estimate state importance and guide predicted return redistribution to be proportional to state importance. The redistributed prediction rewards are then used as an auxiliary target during reward learning. We present results from extensive experiments on complex robot arm manipulation and locomotion tasks and compare against state of the art baselines to demonstrate the impact of Hindsight PRIOR and the importance of addressing the credit assignment problem in reward learning.

**Limitations & Future Work:** Hindsight PRIOR greatly improves PbRL and our qualitative assessment shows that the selected important states are reasonable. However, it relies on the assumption that states that are important to the world model are also important to an arbitrary human. Different humans might attribute importance to different states. Future work will investigate the alignment between the world model's important states and those people focus on when providing preference feedback as well investigation the personalization aspects of important state identification.

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

APPENDIX

## A    DOMAINS FOR EMPIRICAL EVALUATION

We consider six domains of Metaworld Yu et al. (2020) and three domains of DM Control Tunyasu-vunakool et al. (2020) for our empirical evaluation.

For all our experiments we use the internal state representation as the agent state. This includes reward learning, policy learning and world model training. The state features are as described in Yu et al. (2020). Further we follow Park et al. (2022) to utilize the packaged task rewards in MetaWorld for our synthetic oracles.

## B    REWARD ARCHITECTURE FOR PRIOR

We follow Park et al. (2022) reward architecture for Metaworld and DMControl domains.

## C    WORLD MODEL LEARNING

We generally follow the training paradigm suggested in TWM Robine et al. (2023) to learn our forward dynamics model. Furter, we follow the same architecture with minor changes to work with continuous control domains : :

1. TWM is proposed for discrete actions image based atari Brockman et al. (2016) domains, therefore we modify the observation encode layer to take in a 1-D state vector (instead of 3 channel image vector). Additionally, the original TWM makes use of frame-stacking which we do not.

2. We change the state predictor to take continuous valued vector based action space (as in our case) instead of discrete 1-D actions (as in Atari).

Finally, we modify the world model sequence length and memory length to be same as the query length such that we can feed the whole trajectory as an input to the TWM model.

To obtain an attention map from TWM we auto-repressively feed in a trajectory, i.e. first we feed in the first state,action tuple as trajectory of size 1, then the next state, action tuple and so on. At the final step we take the attention vector from the attention layers in the architecture.

### C.1    OBSERVATION MODEL

The observation model in our world model encodes the input state,action tuple. The observation model has two components, an encode and a decoder both of which are multi-layer perceptrons.

The encoder has two hidden layers of size 512 and an output layer of size (32,). The decoder layer has an input size of (32,) followed by two hidden layers of size 512 and the output shape is same as the number of state features. As recommended by Robine et al. (2023) we use SiLU activation in the MLP.

### C.2    LATENT STATE PREDICTOR

We follow Robine et al. (2023) to contruct the world model architecture with minor changes required to adapt to continuous control action space and 1-D state vector. That is, vanilla TWM requires categorical action embeddings (because of the discrete action space) but we do not. Finally, TWM Robine et al. (2023) can operate in different modes with respect to the prediction heads from the latent state predictor such as, next state prediction, reward prediction, and discount prediction. For Hindsight PRIOR we only require the next-state prediction output head.

Similarly TWM supports different input modes i.e. it can take (state,action) tuple and (state, action, reward) tuple. We compute state conditioned hindsight priors and therefore only need the (state, action) as input. Robine et al. (2023) is inconclusive on the need for rewards in the inputs. Moreover,

since PRIOR is learning the reward model to begin with we only use (state, action) as input. Robine et al. (2023) uses a Transformer XL architecture which we borrow as is.

## C.3 Training World Model

We share the replay buffer of the agent with the world model. Similar to PbRL paradigm, the world model first gets access to a bank of state transitions during PbRL's pre-training step. Once the pretraining step is complete the world model is trained on this data (to get the observation model). After this pretraining step only the dynamics model is trained on incoming data every $j$th step of PbRL loop. The world model is trained on its bank of transitions as suggested by Robine et al. (2023).

## D Policy Evaluation Metrics

Algorithms are evaluated according to their learning curves over the course of policy and reward training along with their normalized returns for Deep Mind Control Suite and normalized success rates for MetaWorld Lee et al. (2021b). An algorithm's normalized return or success rate measures how well the policies trained jointly with a preference-learned reward function recovers the performance of a policy trained on the target reward function. For each episode of policy training, the PbRL or SAC policy's return or success rate is computed and then the PbRL policy is evaluated based on how well it is able to recover "optimal" performance approximated with SAC trained on the ground truth reward. The normalized returns are computed as:

$$\text{normalized returns} = \frac{1}{T} \sum_t \frac{r_\psi(s_t, \pi_\phi^{\hat{r}_\psi}(a_t))}{r_\psi(s_t, \pi_\phi^{\bar{r}_\psi}(a_t))}, \tag{6}$$

where $T$ is the number of policy training steps, $\bar{r}_\psi$ is the target reward function, $\pi_\phi^{\hat{r}_\psi}$ is the policy trained in conjunction the learned reward function, and $\pi_\phi^{\bar{r}_\psi}$ is the policy trained on the target reward function. The normalized success rates are computed as:

$$\text{normalized success rates} = \frac{1}{T} \sum_t \frac{\text{success}(\pi_\phi^{\hat{r}_\psi}(a_t))}{\text{success}(\pi_\phi^{\bar{r}_\psi}(a_t))}, \tag{7}$$

where $\text{success}(\cdot)$ indicates whether action $a_t$ resulted in the policy reaching the goal state.

## E Hyper-parameters

A results in the paper are reported over five random seeds: $[12345, 23456, 34567, 45678, 56789]$.

### E.1 Train Hyper-parameters

This section specifies the hyper-parameters (e.g. learning rate, batch size, etc) used for the experiments and results. The SAC, and PEBBLE experiments all match those used in Haarnoja et al. (2018) and Lee et al. (2021a) respectively. The SAC hyper-parameters are specified in Table 2, the PEBBLE hyper-parameters are given in Table 3, the hyper-parameters used to train on with Hindsight PRIOR are in Table 4, and finally the hyperparameters used for training our world model in Table 5. Table 5 only mentions the hyper-parameters that we change in the prescribed Robine et al. (2023) configuration.

Table 2: Training hyper-parameters for SAC (Haarnoja et al., 2018).

| HYPER-PARAMETER | VALUE |
| --- | --- |
| Learning rate | 1e-3 (cheetah), 5e-4 (walker), 1e-4 (quadruped), 3e-4 (Meta-World) |
| Batch size | 512 (DMC), 1024 (Meta-World) |
| Total timesteps | 500k, 1M (quadruped, sweep into) |
| Optimizer | Adam (Kingma & Ba, 2015) |
| Critic EMA $\tau$ | 5e-3 |
| Critic target update freq. | 2 |
| $(\mathcal{B}_1, \mathcal{B}_2)$ | $(0.9, 0.999)$ |
| Initial Temperature | 0.1 |
| Discount $\gamma$ | 0.99 |

Table 3: PEBBLE hyper-parameters (Lee et al., 2021a).

| HYPER-PARAMETER | VALUE |
| --- | --- |
| Learning rate PEBBLE | 3e-4 (Metaworld), 5e-4 (Walker, Cheetah), 1e-4 (Quadruped) |
| Optimizer | Adam (Kingma & Ba, 2015) |
| Segment length $l$ | 50 |
| Feedback amount / number queries ($M$) | 1000/100, 100/10 (DMC) 10000/50, 4000/20, 2000/25, 400/10 (Meta-World) |
| Steps between queries ($K$) | 20000 (walker, cheetah), 30000 (quadruped), 5000 (Meta-World) |

Table 4: Hindsight PRIOR hyper-parameters.

| HYPER-PARAMETER | VALUE |
| --- | --- |
| $\lambda$ | 1000 (MetaWorld), 5 (DMC) |
| update frequency $j$ | 2000 steps |

Table 5: World Model hyper-parameters in Hindsight PRIOR

| HYPER-PARAMETER | VALUE |
| --- | --- |
| world model sequence length | 50 |
| world model memory length | 50 |
| world model batch size | 100 |

## F  NORMALIZED RETURNS AND SUCCESS RATES BY TASK

The mean normalized scores for each algorithm on each task are given for MetaWorld in Table 6 and DMC in Table 7.

A two-tailed paired t-test with dependent means (significant at p ¡ .05) was performed over the normalized returns and success rates to determine that Hindsight PRIOR's performance gains are statistically significant over:

1. **MetaWorld**: PEBBLE ($t = -3.92$, $p = 0.006$), SURF ($t = -2.85$, $p = 0.025$), RUNE ($t = -5.39$, $p = 0.001$), and MRN ($t = -4.91$, $p = 0.002$)

2. **DMC**: PEBBLE ($t = -3.47$, $p = 0.00843$), SURF ($t = -2.52$, $p = .03541$), RUNE ($t = -7.745967$, $p = 0.00006$), and MRN ($t = -2.392232$, $p = 0.0437$)

Table 6: Normalized Success Rate for MetaWorld domains

| Env/Algorithm | PRIOR | PEBBLE | SURF | RUNE | MRN |
|---|---|---|---|---|---|
| **Window Open** | **0.55** | 0.40 | 0.45 | 0.35 | 0.43 |
| **Door Open** | **0.65** | 0.62 | 0.55 | 0.53 | 0.53 |
| **Drawer Open** | **0.68** | 0.59 | 0.59 | 0.59 | 0.53 |
| **Sweep Into** | **0.69** | 0.51 | 0.54 | 0.52 | 0.62 |
| **Button Press** | **0.69** | 0.65 | 0.65 | 0.67 | 0.65 |
| **Door Unlock** | **0.68** | 0.62 | 0.64 | 0.53 | 0.54 |

Table 7: Normalized Returns for DM Control domains

| Env/Algorithm | PRIOR | PEBBLE | SURF | RUNE | MRN |
|---|---|---|---|---|---|
| **Walker Walk** | **0.60** | 0.46 | 0.66 | 0.47 | 0.59 |
| **Cheetah Run** | **0.51** | 0.33 | 0.36 | 0.39 | 0.46 |
| **Quadruped Walk** | **0.65** | 0.66 | 0.64 | 0.60 | 0.54 |

## G  ADAPTED BASELINES FOR PBRL

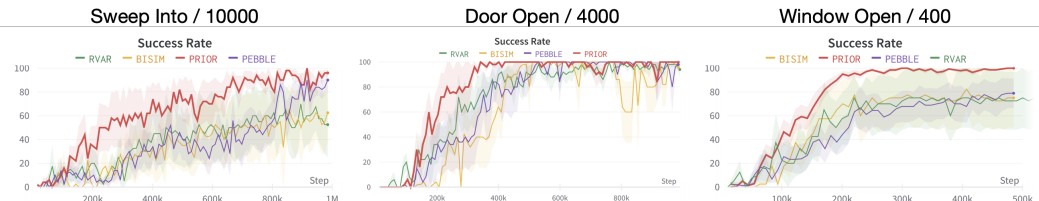

Figure 5: Learning curves of PRIOR, BISIM, RVAR and baseline PEBBLE

### G.1  UNINFORMED RETURN REDISTRIBUTION (RVAR)

Inspired from Ren et al. (2021) we consider the Uninformed return redistribution baseline that essentially has reward targets as the mean reward in the trjaectory, i.e. $r_{target} = \frac{G}{|\tau|}$. This essentially reduces the variance of the trajectory rewards (hence the name RVAR). As discussed previously, RVAR is marginally better than PEBBLE and PRIOR is superior to such an uninformed redistribution technique. (See Fig. 5).

### G.2  BISIMULATION METRIC : BISIM

To incorporate the bisimulation metrics into PbRL, we use the following bisimulation metric loss:

$$\mathcal{J}_{bisim}(\psi) = (||z_i - z_j||_1 - |r_i - r_j| - \gamma||z_i^{'(z_i,a_i)} - z_j^{'(z_j,a_j)})^2, \tag{8}$$

adapted from Equation 4 in Kemertas & Aumentado-Armstrong (2021). We use the reward model to provide the predicted rewards $r_i, r_j$ required by bisim loss. Further we use the penultimate layer as the embedding layer to obtain $z_i, z_j$. Next, we add an additional head from the penultimate layer that predicts the next state embedding to obtain $z_i', z_j'$ as next state predictors. Finally, we reuse the trajectory buffer (as used by Hindsight PRIOR) to obtain the states on which we compute the bisim-target distance and finally optimize the loss as above. We verify that the BISIM adapted baseline offers only marginal improvements over baseline PEBBLE as given in Fig. 5.

### G.3 NORMALIZED REDISTRIBUTION PRIOR (NRP)

Readers may question whether the raw attention values extracted from the forward dynamics prediction task are the best choice or certain post processing may further improve PRIOR's performance. We construct a variant of PRIOR referred to as PRIOR-NRP where we perform a min-max normalization of the attention vector $\alpha$. That is :

$$\hat{\alpha}_i = softmax(\frac{\alpha_i - \alpha_{min}}{\alpha_{max} - \alpha_{min}}) \tag{9}$$

The above equation amplifies the attention values thereby preventing the reward targets to become uniform. Fig. 6 shows that default PRIOR (without any postprocessing of attention) performs well and NRP like post-processing is not required.

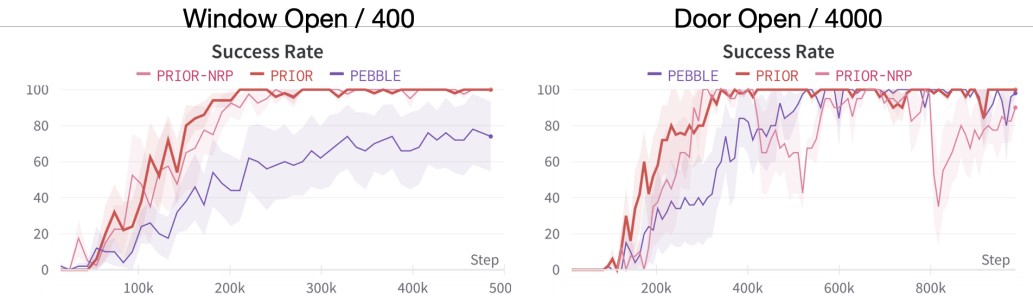

Figure 6: Learning curves of PRIOR, PRIOR-NRP and baseline PEBBLE

## H LONG TRAJECTORIES

Figure 7 shows the performance of PRIOR compared to PEBBLE when the trajectory length is 4x (200). While the Transformer XL architecture is already known for scaling to much longer trajectory lengths Robine et al. (2023) our experiments conclude that this is indeed the case for the challenging continuous control domains.

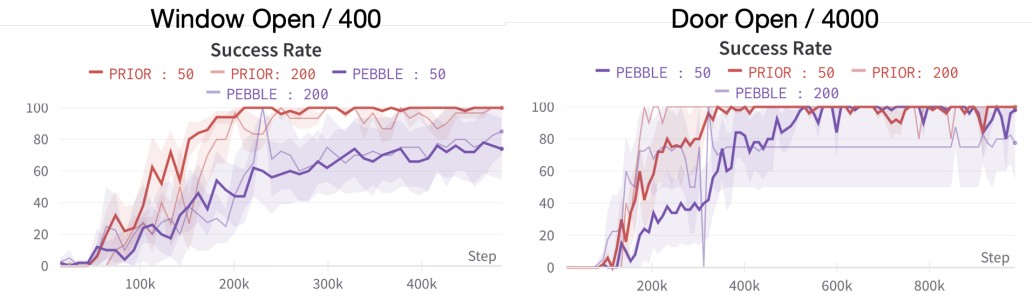

Figure 7: Learning curves of PRIOR and PEBBLE on query segment lengths of 50 and 200.

## I   QUALITATIVE STATE IMPORTANCE ASSESSMENT

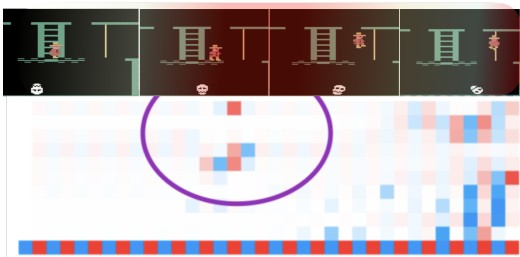

Figure 8: Attention analysis on Montezuma's Revenge. The plot shows a situation where the agent (red animated character) attempts to jump from the green platform towards the rope. Each row represents a layer in the Transformer Model and the columns represent the time step where left most column is time t-T and right most column is the present state of the agent. In hindsight, we compute this attention map to obtain the attention over the past states (given as blue cells) and attention over past actions (given as red cells). Note that the map begins with a blue cell (as $s_0, a_0, a_1 \cdots$). We highlight that the agent attends to past states which corresponds to point of launch which can be considered as an important summary state for the complete trajectory of jumping from the platform to the rope.

We qualitatively evaluate the states identified by the attention weights $\alpha$ as important for a given trajectory. We evaluate whether the attention map $\mathcal{A}$ salience over a trajectory can capture critical events. We conduct experiments on Atari (Montezuma's Revenge) and Metaworld (Window-Open, Door Open) domains, and seek to answer whether the world model attends to states in the complete history (even for Markovian transitions) and whether that correlates with "critical" events (similar to how Kim et al. (2023) describes it). From figure 8 we can see that the forward model does attend to past states and actions. Moreover, upon closer inspection by executing known maneuvers, such as jumping across the platform to the rope in Montezuma's Revenge, we find that the agent is attending to past critical events like the point of launch. We do not expect attention over states in history to be aligned with human's reward function as human may have arbitrary preference unknown to the dynamics model. However, we do expect the attention to be aligned with "critical events" loosely defined as apriori states enough to summarize a trajectory. To further investigate this we force the reward model to predict rewards aligned with the PRIOR reward targets by very high value $\lambda_{prior}$ in equation 5 and find that the reward model is unable to learn preference-relevant reward model at all. This shows that PRIOR reward targets are not task specific but contain salience information on states which can be considered "critical" in the sampled trajectory.

