# OpenReview forum: "Hindsight PRIORs for Reward Learning from Human Preferences"
_ICLR.cc/2024/Conference — ICLR 2024 poster_

### Official Review · Reviewer_7k1k · 2023-10-26

**Soundness:** 3 good
**Presentation:** 3 good
**Contribution:** 3 good
**Rating:** 8
**Confidence:** 3

**Summary:**

This paper presents a new credit assignment strategy for efficiently learning a reward function in preference-based reinforcement learning (PbRL). The preference-based reward model relies on the well-known Bradley-Terry model, and the basic loss consists of the standard cross-entropy loss between the predicted and the true preference labels. The novel contribution of this paper is an additional loss term that redistributes the expected discounted return to each state-action pair in a particular manner. Specifically, a transformer-based forward dynamics model is learned as an auxiliary task, and the expected discounted return is redistributed to be proportional to the attention weights of the state-action pairs. This additional loss term serves as the prior for reward learning, and the authors hypothesize that it leads to sample efficiency and overall policy performance. The empirical evaluation on Deep Mind Control (DMC) and MetaWorld control tasks suggest positive results.

**Strengths:**

A major strength of the paper is that the proposed framework is general in the sense that it can be applied to different baseline PbRL algorithms to improve sample efficiency and performance.
In the beginning it was not intuitive to believe that the return should be redistributed according to attention weights of the forward dynamics, but the explanation provided in Appendix I is somewhat convincing that the attention weights are reflective of critical events that summarize the whole trajectory, and thus it is those critical state-action pairs that contribute the most to the success/failure of the trajectory.

**Weaknesses:**

The authors propose an intriguing approach to boost the performance of PbRL, but some of the empirical results presented in Section 5 reveal some weaknesses. In particular, in the Drawer Open / 4000 task in Figure 2, the success rate of the proposed approach plateaus after 400k steps and is eventually surpassed by some other baselines. This might imply that the hindsight prior eventually hurts the performance of the learned policy as more preference labels become available. Similarly, in Section 5.3 the authors find that a large coefficient on the prior loss leads to a collapse of the learned policy. Those observations indicate that such a prior is assistive of policy learning only up to a certain point (e.g. relatively small preference data or small prior coefficient), and it may eventually hurt the performance if we exceed those bars.

In terms of the presentation of the paper, there is much room for improvement. First, some citations and references are missing and appearing as “?” or “??”. Second, some performance plots in Section 5 have too many curves of similar colors and widths, making it difficult to extract information from them. Specifically, I suggest that the authors use different line styles for Figure 2 (in particular for SAC since it’s an oracular baseline). The left two figures of Figure 3 are also hard to read as there are 8 line plots on each tiny figure. Third, the explanation of hindsight prior is confusing and needs elaboration/clarification. (a) The transformer uses H attention heads, but H does not appear in the definition of the attention matrix A. (b) Equation (4) uses the notation $R_{target}$ and $\hat{R}$, but it is unclear how it is related to $\hat{\mathbf{r}}_{\psi}$ and $\mathbf{r}_{target}$ in the line above.

**Questions:**

1) In the explanation of equation (3), the authors define H as the entropy. Do you mean “cross entropy” there?


2) A recent study [1] finds that reporting the mean and variance for performance evaluation of RL policies is insufficient, and suggests reporting confidence intervals or performance profiles as more objective measures. Have you considered them at all instead of the t-test?


[1] Agarwal, Rishabh, Max Schwarzer, Pablo Samuel Castro, Aaron C. Courville, and Marc Bellemare. "Deep reinforcement learning at the edge of the statistical precipice." Advances in neural information processing systems 34 (2021): 29304-29320.

---

> ### Author Response · Authors · 2023-11-16
> **Response to Reviewer7k1k**
>
> We thank our reviewer for their recommendation to accept our work at ICLR. We are happy and completely agree with our reviewer that one of the main features of Hindsight PRIOR is its generality to be extended to other domains of interest.
>
> On the performance plateau on Drawer Open domain :
> We showcase all of our experiments on 5 seeds. Upon inspecting the performance of individual seeds, we find that the performance drop is mainly due to a single seed <Seed value>. We would like to highlight that we strictly follow prior works for setting up our experiments (i.e. number of seeds etc). Further, while our reviewer presents an interesting hypothesis, the other experiments (8 of 9 domains) do not seem to “plateau”. However, our reviewer is correct in pointing out that PRIOR-obsession can be harmful, but we make that point to highlight the insufficiency of the PRIORs to capture human preference and the need of human feedback data. We believe that our reviewer’s comments indicate the importance of balancing the loss terms especially as the training progresses, which is an important concern for much of Machine Learning that presents auxiliary learning objectives.
>
> On the explanation to H term :
> In Equation 3, H is the entropy term used by TWM (their Equation 4) to learn environment dynamics by encouraging similarity between the predicted latent next state and the true latent next state. They overload the notation of H to mean entropy (for entropy regularization) and cross entropy for latent state predictor. We borrow the notation from TWM work for consistency with their work. This objective is separate from the cross-entropy objective used to learn the reward function in PbRL.
>
> TMW paper: Robine, J., Höftmann, M., Uelwer, T., & Harmeling, S. (2022, December). Transformer-based World Models Are Happy With 100k Interactions. In International Conference on Representation Learning 2023.
>
> On our evaluation protocol :
>
> We thank our reviewer for pointing out the study by Agarwal et al. We would be happy to report confidence intervals in the final version of our work to maintain high quality of reproducibility and reporting of results. Our choice of current evaluation is motivated by high standards of reproducibility as we follow existing works PEBBLE, SURF for the evaluation protocol. The study [1] suggests the need for CI for RL where there exists a large inconsistency in evaluation protocols and how CIs can help. Since PbRL is relatively nascent compared to the large body of RL works, we chose to follow existing state of the art methods for consistent evaluation (in terms of plotting the metrics, hyperparameters, choice of domains etc.). Moreover, the benefits of Hindsight PRIOR can be well understood and appreciated with the current evaluation protocol and given the stark difference in performance our conclusions should remain unchanged. Finally, we agree with our reviewer that PbRL community should also report CIs and we will be happy to include it in our manuscript/appendix upon acceptance of our work.
>
> We are thankful to our reviewer for helpful comments on improving the presentation of the paper and we have made respective changes based on our reviewer’s comments.

---

> ### Comment · Reviewer_7k1k · 2023-11-20
> **Response to Authors**
>
> Thank you for your comments. I agree with authors on the performance of the PRIORs and the evaluation protocol.
>
> Regarding the word "entropy," the authors of the TWM paper indeed overloads the notation $H$  but does use the term "cross entropy" where appropriate (see their explanations around equations (3) and (4) in the TWM paper). Entropy, by definition, depends only on a single distribution, whereas cross entropy depends on two distributions. The distinction here is quite clear and should not be confused. Please make sure in your manuscript to distinguish the two words, even if you overload the notation. And if you choose to overload the notation, please clarify so in the paper.
>
> Please let me know when your updated manuscript is available on OpenReview. Thanks again for your hard work and comments.

---

> > ### Author Response · Authors · 2023-11-23
> > **Response to our reviewer**
> >
> > We thank our reviewer for consider our rebuttal. We are extremely happy that our reviewer agrees with our comments on performance and the evaluation protocol and all of their concerns were addressed through our discussion.
> >
> > We have taken our reviewer's suggestions to clarify H as "cross-entropy", and updated our equation (4) to use a more consistent notation. Furthermore, we have improved the figure captions to be more helpful to our readers in parsing figure 3 as suggested by our reviewer.
> >
> > Finally, we thank our reviewer for their time, effort and constructive suggestions to our work. We hope that our reviewer is more confident in their assessment of our work after our discussion and that they can reconsider their evaluation (rating/confidence) post our discussion.

---

### Official Review · Reviewer_gM8u · 2023-10-31

**Soundness:** 3 good
**Presentation:** 3 good
**Contribution:** 1 poor
**Rating:** 5
**Confidence:** 3

**Summary:**

This paper addresses the credit assignment problem in preference-based Reinforcement Learning (PbRL) algorithms. Given sparse feedback, it is challenging to determine where rewards should be assigned in a trajectory, i.e., which states are significant. The proposed solution combines the classical PbRL algorithm PEBBLE with a prior obtained from a world model. This approach assumes that states receiving high attention in the world model are likely to be rewarding, assigning them higher weight when estimating the reward function. The algorithm is evaluated on simulated problems from the DMC suite and MetaWorld control.

**Strengths:**

The issue of preference-based RL and the credit assignment problem is highly relevant, particularly considering the need for numerous samples to accurately estimate the reward function.

The idea of utilizing the attention layers of the learned world model to identify rewarding state-action pairs is creative and seemingly novel, offering a straightforward but effective solution.

The approach outperforms other methods, demonstrating its effectiveness in comparison.

The paper is well-written and presents its content in an understandable manner.

**Weaknesses:**

The primary limitation of this work, as acknowledged in the paper, is its reliance on the assumption that states deemed important by the world model are also significant for reward design. While this insight is valuable, the contribution of the paper might be relatively modest, given that the primary novelty lies in a straightforward implementation of this assumption and its evaluation. With that, the quality of the contribution may not fully meet the requirements for acceptance at ICLR.

The paper would benefit from additional work to clarify the extent to which the learned attention in world models aids in task characterization for interpretability and transferability, as these are key applications of reward learning (building on Q3 in the paper).

__Typos:__

Page 2, "Learning World Models": "us it to" should be corrected to "use it to."

Page 4, "a local minima" (plural) should be corrected to "a local minimum."

**Questions:**

How do you expect the performance of your algorithm to compare with more sample-efficient algorithms, such as few-shot preference learning [1]?

[1] Hejna, Joey & Sadigh, Dorsa (2023). Few-shot preference learning for human-in-the-loop rl. In Conference on Robot Learning (pp. 2014-2025). PMLR.

---

> ### Author Response · Authors · 2023-11-15
> **Response to Reviewer gM8u**
>
> We thank our reviewer for their insightful comments and appreciate the feedback that the paper is “creative, novel, straightforward yet effective” and the manuscript well-written.
>
> On the quality of our contribution :
> We argue that simple ideas and straightforward and intuitive implementation, as appreciated by our reviewer (and other reviewers), are strong indicators of extensibility and reusability of the work by the community. For example, past works gaining traction in PbRL community like PEBBLE (realizes a way to use SAC in PbRL), SURF (realizes a way to use a simple semi-supervised objective), etc. also offer a simple, intuitive and effective novelty that can be easily evaluated, extended and leveraged in future works.
>
> We present Hindsight PRIOR as a combination of several key contributions including our insight into the usefulness of world model attention for reward learning (as appreciated by all our reviewers), a no-cost-to-human way of obtaining priors through our insight, in highlighting the limitation of dependence on the standard cross-entropy loss, and a “creative” method of incorporating computed PRIORs into the reward learning process via predicted return redistribution. Finally, as noted by all our reviewers, we showcase the advantage of our work over several baselines and perform a thorough evaluation on domains interesting to the PbRL community.
>
> On Interpretability, Transferability and multi-task PbRL extensions :
> While our reviewer raises an interesting point on extended evaluation of Q3, we argue that it should be considered as an “additional” evaluation, or a next step, rather than necessary for this paper. While interpretability is an important area for Machine Learning, measuring the interpretability of a learned reward function on tasks such as locomotion and manipulation is an open area of research. Similarly, while transferability of rewards is an important area to consider, we follow our baselines and focus on addressing the challenge of sample inefficiency in a single domain. When transferability is an important requirement (such as, when optimizing across a suite of tasks and human preferences can also transfer) the PbRL setting needs to be augmented. That is, the aim becomes to exploit the sub-structure across tasks within the domain for improvement and we appreciate the reviewer highlighting that our method may help with this. In [1] the work assumes knowledge of a suite of tasks and “extends meta learning framework over preference learning”. Hindsight PRIOR can certainly be leveraged in such a setting as we leverage a world model where the challenge would be to learn a world model (shared across tasks) to obtain PRIORs. Our reviewer presents a great natural extension of Hindsight PRIOR for multi-task preference setups, as our work exploits the dynamics information within a task and would be extended to exploiting dynamics sub-structure across tasks. However, due to our experimental setup (information available for single task v/s multi-task) and goal (leveraging preference feedback on current task v/s leveraging task preference similarity across tasks) versus their’s a direct comparison is not suitable.
>
> We believe the above addresses you questions/concerns. Please let us know if our response missed any part of your questions/concerns.

---

> > ### Comment · Reviewer_gM8u · 2023-11-22
> >
> > Thank you for your comments. I still believe that the contribution, i.e., using the attention variables from the world model straight-forwardly as a prior for PbRL without theoretical contribution or reward design analysis, is rather weak for an ICLR paper.

---

> > > ### Author Response · Authors · 2023-11-23
> > > **Response to our reviewer**
> > >
> > > We would like to thank our reviewer for considering our rebuttal. We hope we were able to address remaining concerns they had.
> > >
> > > Finally, we would like to highlight that we do perform an analysis of the alignment of attention map & the task as in Appendix I. Furthermore, we would like to reiterate that our contribution spans our insight on the usefulness of model attention, our proposal to use them as "PRIORs" including a no-cost-to-human way of obtaining these priors (through TWM world model's forward state prediction task), and folding them into the reward learning process (predicted return redistribution). Further, we highlight the limitation of dependence on standard cross-entropy loss. Similar to past pivotal works in PbRL like SURF and PEBBLE we show strong empirical gains of using Hindsight PRIORs on several domains of interest to the community and across interesting baselines.
> > >
> > > We are happy that our reviewer appreciates our problem highly relevant,  that our solution offers simplicity and effectiveness, and our method "creative and novel". We hope that our reviewer can reconsider their evaluation of our work based on our discussion.
> > >
> > > We have taken our reviewer's suggestions and updated our manuscript to correct for missing citations and references, and corrected the typos.

---

### Official Review · Reviewer_4UMz · 2023-11-01

**Soundness:** 3 good
**Presentation:** 3 good
**Contribution:** 3 good
**Rating:** 6
**Confidence:** 4

**Summary:**

This work presents Hindsight PRIOR, a novel technique to guide credit assignment to improve reward inference in Preference-based Reinforcement Learning. The key contribution in this paper is the utilization of attention weights from a transformer-based world model to estimate state importance and the formulation of return redistribution to be proportional to the attention-deduced state importance. The authors present information regarding related work, their approach, and an empirical evaluation in the Deep Mind Control and MetaWorld Control Suites. The results with a synthetic labeler are positive, displaying PRIOR achieves high success across a variety of tasks.

**Strengths:**

+ The proposed method is an improvement to PbRL frameworks. Given the references in the paper and how humans may utilize attention similarly to transformer models, utilizing attention weights to redistribute return may improve preference-based reinforcement learning with  end-users.
+  This paper is well-written and contains sufficient detail to understand the proposed approach.
+ The evaluation is extensive, and touches on several important questions beyond simple performance.

**Weaknesses:**

- It would be beneficial to note exactly how many trajectory labels such a framework requires. This would help detail whether such a framework would be feasible with actual end-users. Further, including actual tests utilizing this framework with human end-users would provide further evidence that PRIOR works well.
- Along this thread, it seems the simultaneous learning of a highly parameterized world model and reward model is accomplished faster than other works that simply inferring a reward model, as shown by the sample-efficiency in policy learning. Could you comment on why this is the case? I'm unsure if this relates to a paragraph on page 2 referencing the choice of architecture of the reward network.
- As PRIOR utilizes PEBBLE as its backbone algorithm, this should be touched on in the related work.
-  In the evaluation, there are several references that are not labeled correctly and lead to ??. As several of these baselines are not referenced or explained previously, it leads to confusion regarding the results.
- Could you provide justification on why the attention coefficient for the state and action should be equally waited within the \alpha coefficient?

**Questions:**

Please address the weaknesses above.

---

> ### Author Response · Authors · 2023-11-15
> **Response to Reviewer 4UMz**
>
> We would like to thank the reviewer for their insightful comments, and are happy that they found our work well-written, and our evaluation extensive and answering questions beyond performance.
>
> On the amount of trajectory labels required by PRIOR :
> We would like to highlight that the number of steps in Figures 2, 3, and 4 are the number of policy steps (x-axis). Following the backbone PEBBLE, at each step policy learning (SAC update) happens, every few RL steps the reward learning step is executed (for example, 5000 steps for Metaworld) along with query for new human feedback data, and every few RL steps the world model update happens (2000 steps for all experiments). The total amount of feedback is given by the end of policy training. Additional details on reward update schedules are available in Appendix E.1 for our reviewers. Therefore, the world model training steps are “in-between” RL episodes. Since the feedback schedule is based on the RL training steps, the plots highlight feedback efficiency and performance improvements shown by PRIOR. However, the training of the world model indeed poses an additional compute requirement but the additional world model training’s wall-clock time is insignificant compared to other costs like RL training, human feedback time (in a real world setup) etc. We recall that our choice of the world model backbone (TWM) is designed to be sample efficient as is shown by authors of TWM work on Atari benchmark. Moreover, such a cost can be easily reduced as the world model can be trained parallelly with RL training (and the two processes need to be synced only at certain episodes). Finally, we implemented a “serialized” version which performs each step i.e. policy learning, reward update, world model update in a sequential manner and found that the wall-clock time difference was no more than an hour when compared to baseline PEBBLE.
>
> Intuition on balancing the auxiliary loss term :
> An important feature of Hindsight PRIOR is that the loss coefficient was set by intuition, as the reviewer correctly recalls, to make the two losses “equally weighted”. While better hyperparameter sweep strategies can be used to potentially improve the performance, our intuition was simple that, in the absence of evidence to the contrary, both the loss components should contribute equally to the final optimization and yield a more balanced training. Based on our reviewers suggestion, future research can indeed look at the benefits of dynamically balancing the auxiliary loss.
>
> On User Study :
> We thank the reviewer for their suggestion on a human subject study with PRIOR. While additional experiments (as our reviewer highlights) can be helpful in understanding PRIOR’s gains, we believe that our current set of experiments thoroughly investigate Hindsight PRIOR. Specifically, we conduct experiments using “mistake oracle” with high degree of mistake value (i.e. oracle feedback is flipped with some $\epsilon$) and showcase the robustness of PRIOR over baseline methods to different frequencies of mistake. While a user study may help us find the exact number of feedback, our aim is to present PRIOR as a general approach and highlight the significant performance improvements over baselines.
>
> We apologize for the missing references which will be corrected during the rebuttal phase (along with other constructive suggestions on manuscript presentation from all our reviewers).
>
> We believe the above addresses you questions/concerns. Please let us know if our response missed any part of your questions/concerns.

---

> > ### Comment · Reviewer_4UMz · 2023-11-21
> > **Rebuttal Response**
> >
> > Thank you for your response! Your responses have addressed my questions and concerns.

---

> > > ### Author Response · Authors · 2023-11-23
> > > **Response to our Reviewer**
> > >
> > > We are extremely happy that all our reviewer's concerns were addressed. We hope that our reviewer can reconsider their evaluation rating / confidence based on our discussion.
> > >
> > > We have taken into account our reviewer's suggestions on correcting missing citations and will post the revised draft pdf.

---

### Author Response · Authors · 2023-11-23
**Response to all our reviewers**

We would like to thank all our reviewers for their time, effort and wonderful and constructive suggestions to our work.

We are extremely happy that we were able to address all our reviewer's concerns (4UMz, 7k1k). We are also very happy that our reviewers found our problem highly relevant and important, manuscript well written, our evaluation extensive, our method creative, novel and general (in that it can be applied to other settings); as a good contribution should.

We have posted a revised manuscript with changes suggested by our reviewers as follows :
1. We have corrected for missing citations and references. (suggested by all our reveiwers)
2. We have fixed typos (suggested by gM8u)
3. We have updated our equation 4 to use a more consistent notation and clarification on "H" as cross-entropy (suggested by 7k1k)
4. We have added improved caption to figure 3 for better readability for readers. (suggested by 7k1k).

---

### Meta-Review · Area_Chair_MQsb · 2023-12-05

**Metareview:**

**Summary**: This paper focuses on guiding credit assignment to improve preference-based reward learning. The core contribution is the use of attention weights from a transformer-based world model, with the assumption that states receiving high attention in the world model are likely to be rewarding, so they should have higher weight when estimating the reward function. The algorithm is evaluated in simulation with a synthetic labeler in the Deep Mind Control and MetaWorld Control Suites.

**Strengths:**
- Using attention weights from the learned world model to identify rewarding state-action pairs is an interesting and creative idea.
- The approach outperforms other (compelling) baselines.
- The paper is well-written and clear.
- The evaluation is extensive and touches on several important questions beyond simple performance.

**Weaknesses**:
- The method is only evaluated with synthetic human feedback. A user study with real end-users testing out the approach would've been more compelling. That said, I really appreciated the 10%, 20%, and 40% noise injection experiments to validate that perfect labeling isn't required.
- The paper plots are really hard to read. Any future version of the paper should improve them for readability.

Overall, I think the paper presents a neat idea for preference-based reward learning. There are some clarity issues to be addressed, but the extensive experiments, investigative approach to experimental analysis, and compelling baseline comparisons merit publication in my opinion.

**Justification For Why Not Higher Score:**

All the results are synthetic at the end of the day, and the idea is still simple (but effective).

**Justification For Why Not Lower Score:**

I think the idea is neat and well evaluated, and the research community would benefit from it. Attention is great: it's not too cumbersome to engineer but positively impacts performance, so I think the paper should be accepted.

---

### Decision · Program_Chairs · 2024-01-16

Accept (poster)